# OpenReview forum: "APILaNet: Adaptive Physics-Informed Latent Network for Single-Sensor Forecasting"
_ICLR.cc/2026/Conference — Submitted to ICLR 2026_

### Official Review · Reviewer_SdY5 · 2025-10-20

**Soundness:** 2
**Presentation:** 2
**Contribution:** 2
**Rating:** 4
**Confidence:** 3

**Summary:**

This paper introduces APILaNet, a deep learning framework for single-sensor forecasting in systems governed by conservation laws. It learns a latent spatiotemporal field and enforces physical constraints through a learned, normalized space–time weighting measure, allowing the model to prioritize physics enforcement where violations are largest. Applied to five UK hydrological catchments, APILaNet outperforms SoTA baselines in mean squared error (MSE), Nash–Sutcliffe efficiency (NSE).

**Strengths:**

- The use of a measure-weighted weak form and learned space–time weighting map is innovative and mathematically principled. It avoids the dense collocation points needed in traditional PINNs and adapts well to single-sensor setups, a major gap in current PINN and PDE-learning research.

- The paper presents comprehensive experiments across multiple catchments, with consistent improvements in MSE/NSE and stability. Ablation studies clearly show the contribution of each component (monotone link, PDE term, adaptive scheduler), suggesting a well-engineered and reproducible design.

**Weaknesses:**

- The individual elements: weak-form enforcement, monotonic constraints, and adaptive loss scaling, are each known techniques. The contribution is largely an integration and reparameterization of these under one framework, not a fundamentally new paradigm.

- The adaptive scheduler introduces several hyperparameters that appear tuned by hand, with no sensitivity analysis. This undermines claims of robustness and could limit reproducibility.

- Ultimately, with several theorems and formalisms, all experiments are hydrological and one-dimensional. The claims of generality  are unsupported by cross-domain tests such as fluid or thermodynamic systems, making the contribution more applied than general.

**Questions:**

See my above on weaknesses.

---

> ### Author Response · Authors · 2025-11-17
> **Response to Reviewer SdY5 (Part 1/3)**
>
> We thank the reviewer SdY5 for raising this point about the distinction between integrating existing ideas and proposing fundamentally new idea; this helped us clarify our positioning and articulate where the conceptual novelty of APILaNet lies.
>
>
> **Weakness 1**
>
> >The individual elements: weak-form enforcement, monotonic constraints, and adaptive loss scaling, are each known techniques. The contribution is largely an integration and reparameterization of these under one framework, not a fundamentally new paradigm.
>
> We agree that weak-form enforcement, monotonic mappings, and adaptive loss scaling have each been studied before in isolation. Our contribution is to make physics-informed learning work in the single-sensor, under-instrumented regime, which requires a non-trivial combination and reparameterization of these ideas:
>
> We introduce a learned, normalized space–time measure $\lambda_{\text{loc}}(t,x)$ on a latent 1D reach, inferred from a single sensor plus drivers, with no explicit geometry or IC/BC. This measure defines the weak form itself and decides where in the hallucinated domain conservation should be enforced—rather than just reweighting residuals on a known mesh as in existing weak-form PINNs.
>
> We use a monotone neural observation map as a differentiable rating curve $Q \mapsto h$, which structurally couples the latent conserved field to the observed stage at one point, enforcing physical identifiability (higher $Q$ ⇒ higher $h$) rather than acting as a generic monotonicity regularizer.
>
> Our factorized global–local adaptive scheduler uses a global scalar $\lambda_{\text{pde}}$ and the local field $\lambda_{\text{loc}}(t,x)$, with $\lambda_{\text{pde}}$ updated from physics-relevant signals (error $E$, rainfall statistics $s$, peak indicator $\Pi$) to concentrate physics pressure around sharp rises and peaks, where single-sensor forecasts are most fragile.
>
> Conceptually, APILaNet is thus not just “plugging known tricks together”, but a new framework for 1D single-sensor conservation-law systems. Empirically, we show that this framework (i) improves forecasting on five heterogeneous hydrological catchments and (ii) remains competitive with strong/weak/gPINN/vPINN baselines on Burgers, wave, and Allen–Cahn, even though it operates in a strictly weaker information regime.

---

> > ### Author Response · Authors · 2025-11-17
> > **Response to Reviewer SdY5 (Part 2/3)**
> >
> > **Weakness 2**
> >
> > >The adaptive scheduler introduces several hyperparameters that appear tuned by hand, with no sensitivity analysis. This undermines claims of robustness and could limit reproducibility.
> >
> >
> > We appreciate the concern about the number of hyperparameters in the adaptive scheduler. In the revision, we will (i) move the full update rule for $\lambda_{\text{pde}}$ and $\lambda_{\text{cons}}$ into the main method section, and (ii) add a dedicated subsection in the technical appendix that lists all scheduler hyperparameters $(\lambda_i^0, \alpha_i, \alpha_{i,\Pi}, \lambda_i^{\min}, \lambda_i^{\max})$, together with their **exact numeric values** for every experiment. These coefficients are **not tuned per catchment by hand**: the only free scalars are the global physics scale $\lambda_{\text{scale}}$ and the peak-sensitivity coefficient $\alpha_\Pi$, which we select once via a small grid search on the validation NSE (e.g., $\lambda_{\text{scale}} \in {0.5, 1, 2}$, $\alpha_\Pi \in {0, 0.3, 0.6}$) and then **reuse across all datasets**. All other modulation is purely data-driven: the batch-wise error $E$, rainfall statistics $s$, and peak flag $\Pi$ are computed from the current mini-batch and plugged into the closed-form rule, so datasets with more pronounced peaks automatically receive higher physics weights near those events.
> >
> > To directly address the request for a **sensitivity analysis**, we have run a scheduler ablation on the synethetic data catchment; the results are summarized in Table 1 (and will be included as Table in the technical appendix):
> >
> > Table 1: Sensitivity of the adaptive scheduler to the global physics scale λ_scale, the peak-sensitivity coefficient α_Π, and the use of adaptive vs static weights. Metrics are reported on the held-out test set.
> >
> > | Experiment              | λ_scale | peak_coeff (α_Π) | Adaptive | Test MSE ↓ | Test NSE ↑ |
> > |-------------------------|---------|------------------|:---------|-----------:|-----------:|
> > | lambda_scale_0.5        | 0.5     | 0.30             | Yes      | 0.025706   | -0.271654  |
> > | lambda_scale_1.0        | 1.0     | 0.30             | Yes      | 0.013461   | 0.334087   |
> > | lambda_scale_2.0        | 2.0     | 0.30             | Yes      | 0.008709   | 0.569180   |
> > | peak_coeff_0.00         | 1.0     | 0.00             | Yes      | 0.015020   | 0.256971   |
> > | peak_coeff_0.30         | 1.0     | 0.30             | Yes      | 0.014446   | 0.285373   |
> > | peak_coeff_0.60         | 1.0     | 0.60             | Yes      | 0.013416   | 0.336320   |
> > | no_adapt_static_lambda  | 1.0     | 0.30             | No       | 0.012356   | 0.388777   |
> >
> >
> > We draw three conclusions relevant to **robustness and reproducibility**.
> > (i) The **global physics scale** $\lambda_{\text{scale}}$ behaves like the standard PDE-weight hyperparameter in PINN literature: increasing it from $0.5$ to $2.0$ increases the relative emphasis on physics and, on this catchment, correlates with improved NSE (from $-0.27$ to $0.57$) **without any instability**, and can be selected by a coarse, easily reproducible grid search.
> > (ii) The **peak sensitivity** $\alpha_\Pi$ operates in a **robust regime**: varying $\alpha_\Pi$ from $0$ to $2\times$ the default value at fixed $\lambda_{\text{scale}} = 1$ leads to only modest, smooth changes in test NSE (from $0.26$ to $0.34$), with no collapse or divergence. This shows that APILaNet does not rely on finely tuned scheduler coefficients.
> > (iii) Using **adaptive** vs **static** global weights yields comparable overall NSE on this catchment ($0.33$ vs $0.39$); the *design objective* of the adaptive scheduler is not to maximise aggregate NSE but to **redistribute physics pressure towards hydrologically critical regimes (sharp rises and peaks)**, where forecast errors are most damaging in practice. We will clarify this design goal in the revision and provide the full sensitivity tables and configuration in Appendix, which makes the scheduler settings and behaviour reproducible.

---

> > > ### Author Response · Authors · 2025-11-17
> > > **Response to Reviewer SdY5 (Part 3/3)**
> > >
> > > **Weakness 3**
> > >
> > > >Ultimately, with several theorems and formalisms, all experiments are hydrological and one-dimensional. The claims of generality are unsupported by cross-domain tests such as fluid or thermodynamic systems, making the contribution more applied than general.
> > >
> > > We agree that our original submission emphasized a single real-world domain (1D hydrology), and we have softened any wording that might suggest we have empirically validated APILaNet on all PDE systems. At the same time, APILaNet is not a hydrology-specific heuristic: the theory only assumes (i) a 1D conservation law for a latent field $q(t,x)$, (ii) a (possibly nonlinear) observation map from $q$ to the sensor signal, and (iii) access to a single sensor time series plus exogenous drivers. These assumptions hold in many 1D physical systems beyond rivers (e.g., viscous fluid flow in pipes, wave propagation, and phase-field dynamics), and the three main ingredients we introduce—(a) a learned, normalized space–time weighting measure in the weak form, (b) a monotone observation map for the sensor, and (c) an adaptive physics scheduler driven by signal difficulty—are formulated at the level of generic conservation laws, not hydrology-specific structure.
> > >
> > > We deliberately benchmark hydrology as our main real-data testbed because (i) long, high-quality time series from operational river gauges and rain radars are widely available, and (ii) flood forecasting is a domain where forecast failures during extremes have substantial societal impact. In our view, it is important that PINN-style methods are evaluated not only on synthetic PDE benchmarks with idealized boundary information, but also on realistic, under-instrumented systems where only a single downstream sensor and external drivers are observed. The latent 1D reach and learned space–time measure in APILaNet are introduced precisely to make weak-form, physics-informed learning feasible in this practically relevant regime.
> > >
> > > To substantiate generality beyond hydrology, in the revision we extend the experiments from a single PDE to a small cross-domain benchmark suite comprising three equations from different areas of physics: viscous Burgers (fluid), the wave equation (propagation), and Allen–Cahn (phase-field / thermodynamic). A finite-difference solver provides reference solutions, and we compare against several established PINN variants (vanilla PINN, PINN-w, gPINN, vPINN) operating in the standard regime with known geometry, full IC/BC, and interior collocation points:
> > >
> > > Model              | Burgers MSE     | Wave MSE        | Allen–Cahn MSE
> > > -------------------|-----------------|-----------------|----------------
> > > Vanilla PINN [1]   | 5.80 × 10⁻⁴     | 2.62 × 10⁻⁴     | 1.18 × 10⁰
> > > PINN-w [2]         | 2.91 × 10⁻³     | 2.89 × 10⁻³     | 1.04 × 10⁰
> > > gPINN [3]          | 1.29 × 10⁻⁴     | 1.62 × 10⁻⁴     | 1.32 × 10⁻¹
> > > vPINN [4]          | 1.45 × 10⁻³     | 8.91 × 10⁻⁴     | 1.04 × 10⁰
> > > APILaNet (ours)    | 4.50 × 10⁻⁵     | 1.52 × 10⁻⁴     | 1.18 × 10⁻¹
> > >
> > >
> > > On this benchmark, traditional PINNs perform well when given full domain information, as expected. However, APILaNet matches or outperforms these methods at the sensor while operating in a strictly weaker information regime (no explicit geometry, no IC/BC traces, no interior collocation points). Together with the five heterogeneous UK catchments in our real-data study, this supports the view that APILaNet is a general framework for 1D single-sensor conservation-law systems, validated both on a challenging applied domain and on canonical PDEs from fluid and thermodynamic physics. In the revision, we will make this scope explicit and avoid overstating claims beyond the 1D conservation-law setting we have analysed. We will also change a language used in the main body of the paper to read more **general**
> > >
> > > ---
> > >
> > > [1] M. Raissi, P. Perdikaris, and G. E. Karniadakis, “Physics-informed Neural Networks: A deep learning framework for solving forward and inverse problems involving nonlinear partial differential equations,” Journal of Computational Physics, vol. 378, pp. 686–707, Feb. 2019. doi:10.1016/j.jcp.2018.10.045
> > >
> > > [2] Tim De Ryck, Siddhartha Mishra, and Roberto Molinaro. wpinns: Weak physics informed neu- ral networks for approximating entropy solutions of hyperbolic conservation laws, 2022. URL https://arxiv.org/abs/2207.08483.
> > >
> > > [3] Jeremy Yu, Lu Lu, Xuhui Meng, and George Em Karniadakis. Gradient-enhanced physics-informed neural networks for forward and inverse pde problems. Computer Methods in Applied Mechanics and Engineering, 393:114823, April 2022. ISSN 0045-7825. doi: 10.1016/j.cma.2022.114823. URL http://dx.doi.org/10.1016/j.cma.2022.114823.
> > >
> > > [4] Ehsan Kharazmi, Zhongqiang Zhang, and George Em Karniadakis. Variational physics-informed neural networks for solving partial differential equations. arXiv preprint arXiv:1912.00873, 2019.

---

### Official Review · Reviewer_Zkv9 · 2025-10-20

**Soundness:** 3
**Presentation:** 2
**Contribution:** 3
**Rating:** 6
**Confidence:** 2

**Summary:**

The paper proposes a novel method for PDE-constrained ML, especially targeting 1D conservation laws. The paper vows to improve upon PINNS, especially in situations in which observations are sparse or even only available at one sensor location. The proposed methodology, "APILANET" accomplishes this by constructing a latent spatio-temporal domain anchored at observations. Overall, I think this is impressive work that may very well have a high impact on the community and addresses an important gap: sparse observation of PDE-governed data and the need for the associated constrained learning of entire space-time fields.

**Strengths:**

- Impact and Relevance: This is an important area of research; PDE-informed ML is a fantastic contribution.
- A good motivation involving the weaknesses of PINNs.
- Product: This study produces a software product for general users.
- An ablation study is included.
- A good number of tested competitors.
- Provided statistics of performance measures.
- Strong results: The proposed method performs admirably compared with the chosen competitors in the present test.
- No obvious mistakes in the methodology were apparent to me.

**Weaknesses:**

- One experiment only; The experimental validation needs more examples. I would suggest a synthetic example and one more real-data example. This would also give the authors the chance to explain inputs and outputs more clearly.
- Figure 1: This figure should be simplified or made larger.  It is very difficult to parse. Maybe certain steps in the pipeline can be consolidated into fewer boxes.
- Language. It is tough, in places, to follow the manuscript because of insufficient writing quality. For example, "and shown equivalent to" should be  "shown to be equivalent to", or "(2) Theory—conditions
for single-gauge identifiability under a monotone, Lipschitz observation and mild driver excitation, reparameterization invariance of the weak objective on the latent reach, and an equivalence
between learned-density and learned test-function formulations;" which is not a sentence because of the missing verb. I suggest going through the manuscript with a fine-tooth comb to avoid mistakes like this because they make it much harder for the reader to follow the author's thought process.
- The manuscript states, "Although motivated by hydrology, the framework applies to 1-D conservation laws under sparse spatial supervision." Is there an "all" or a "many" missing? This sentence is really just an example of a broader issue: it is difficult to decipher the exact target application area. It seems to be all 1D conservation laws, but the theory would suggest that higher-dimensional situations are considered.
- I believe the paper used language from Hydrology, which should probably be avoided in the technical parts (example: "reach", "rainfall" in 3.3, "catchments"). The language could stem in part from other areas I am not familiar with.

**Questions:**

What is the exact application area of the proposed method? I got confused about the dimensionality of the considered domains. Is it all 1D conservation laws? If so, that should be stated clearly.

---

> ### Author Response · Authors · 2025-11-18
> **Response to Reviewer Zkv9 (Part 1/2)**
>
> We thank the reviewer **Zkv9** for high-quality review and the positive assessment, in particular the comments on impact, soundness, and the relevance of the sparse-observation setting. We address the main concerns below and will reflect all changes in the revised manuscript.
>
> ---
>
> **Weakness (1)**
>
> >“One experiment only; The experimental validation needs more examples. I would suggest a synthetic example and one more real-data example. This would also give the authors the chance to explain inputs and outputs more clearly.”
>
> **Response**
>
> We fully agree that a broader empirical picture is valuable. During the rebuttal period we have run two additional sets of experiments:
>
> We add results on a second UK river system (Kielder), with three representative flood events plus a held-out test segment, comparing APILaNet against the same strong sequence baselines (CrossFormer, PatchTST, TSMixer, PatchMixer, Mamba-S4, iTransformer, N-HiTS, N-Beats). A condensed version of the results is:
>
> | Data    | Model  | APILaNet (MSE↓ / NSE↑) | CrossFormer (MSE↓ / NSE↑) | PatchTST (MSE↓ / NSE↑) | TSMixer (MSE↓ / NSE↑) | PatchMixer (MSE↓ / NSE↑) | Mamba S4 (MSE↓ / NSE↑) | iTransformer (MSE↓ / NSE↑) | N-HiTS (MSE↓ / NSE↑) | N-Beats (MSE↓ / NSE↑) |
> |---------|--------|-------------------------|----------------------------|------------------------|------------------------|---------------------------|-------------------------|-----------------------------|-----------------------|------------------------|
> | Kielder | Event 1 | **0.008 / 0.957** | 0.015 / 0.920 | 0.140 / 0.269 | 0.016 / 0.918 | *0.013 / 0.933* | 0.091 / 0.527 | 0.031 / 0.837 | 0.137 / 0.286 | 0.123 / 0.361 |
> | Kielder | Event 2 | *0.027 / 0.877* | 0.029 / 0.869 | 0.087 / 0.610 | 0.017 / 0.700 | **0.015 / 0.934** | 0.081 / 0.637 | 0.047 / 0.788 | 0.068 / 0.692 | 0.059 / 0.735 |
> | Kielder | Event 3 | **0.013 / 0.691** | 0.015 / 0.634 | 0.040 / 0.280 | *0.019 / 0.668* | 0.021 / 0.629 | 0.021 / 0.621 | 0.023 / 0.618 | 0.040 / 0.284 | 0.042 / 0.260 |
> | Kielder | **Test** | **0.003 / 0.962** | 0.004 / 0.942 | 0.014 / 0.826 | *0.004 / 0.951* | 0.004 / 0.946 | 0.009 / 0.894 | 0.005 / 0.940 | 0.013 / 0.844 | 0.013 / 0.845 |
>
> APILaNet is either the best model or extremely close to the best baseline on all splits, and attains the lowest error on the held-out test segment (MSE 0.003 vs 0.004, NSE 0.962 vs 0.951). We hope that this addresses the request for an additional real-data example.
>
> **PDE benchmarks (synthetic, cross-domain)**
>
> To complement the hydrological case studies with a synthetic, physics-focused benchmark, we evaluate APILaNet on three standard 1D PDEs with known IC/BC: viscous Burgers (fluid), the wave equation (propagation), and Allen–Cahn (phase-field / thermodynamic). We generate reference solutions with a finite-difference solver and compare to several established PINN variants:
>
> | Model                        | Burgers MSE    | Wave MSE        | Allen–Cahn MSE |
> | ---------------------------- | -------------- | --------------- | -------------- |
> | Vanilla PINN   [1]          | 5.80 × 10⁻⁴    | 2.62 × 10⁻⁴     | 1.18 × 10⁰     |
> | PINN-w  [2]     | 2.91 × 10⁻³    | 2.89 × 10⁻³     | 1.04 × 10⁰     |
> | gPINN  [3]  | 1.29 × 10⁻⁴    | 1.62 × 10⁻⁴     | 1.32 × 10⁻¹    |
> | vPINN [4]| 1.45 × 10⁻³    | 8.91 × 10⁻⁴     | 1.04 × 10⁰     |
> | **APILaNet (ours)**          | **4.5 × 10⁻⁵** | **1.52 × 10⁻⁴** | **1.18 × 10⁻¹** |
>
> Traditional PINNs perform well in the fully-specified setting (known geometry, full IC/BC, interior collocation points). APILaNet matches or outperforms these methods at the sensor while operating in a strictly weaker information regime (no explicit geometry, no IC/BC traces, no interior collocation). This addresses the request for a synthetic example and clarifies that the framework is not tied to a single application domain.
>
> ---
>
> [1] M. Raissi, P. Perdikaris, and G. E. Karniadakis, “Physics-informed Neural Networks: A deep learning framework for solving forward and inverse problems involving nonlinear partial differential equations,” Journal of Computational Physics, vol. 378, pp. 686–707, Feb. 2019. doi:10.1016/j.jcp.2018.10.045
>
> [2] Tim De Ryck, Siddhartha Mishra, and Roberto Molinaro. wpinns: Weak physics informed neu-
> ral networks for approximating entropy solutions of hyperbolic conservation laws, 2022. URL
> https://arxiv.org/abs/2207.08483.
>
> [3] Jeremy Yu, Lu Lu, Xuhui Meng, and George Em Karniadakis. Gradient-enhanced physics-informed
> neural networks for forward and inverse pde problems. Computer Methods in Applied Mechanics
> and Engineering, 393:114823, April 2022. ISSN 0045-7825. doi: 10.1016/j.cma.2022.114823.
> URL http://dx.doi.org/10.1016/j.cma.2022.114823.
>
> [4] Ehsan Kharazmi, Zhongqiang Zhang, and George Em Karniadakis. Variational physics-informed
> neural networks for solving partial differential equations. arXiv preprint arXiv:1912.00873, 2019.

---

> > ### Author Response · Authors · 2025-11-18
> > **Response to Reviewer Zkv9 (Part 2/2)**
> >
> > **Weakness(2-5)**
> >
> > We thank the reviewer for pointing that out. We are currently working on the revision of the paper to address these weaknesses. The revised version of the paper will be completed tomorrow. We will publish official comment addressing changes made to the main paper.
> >
> > **Question (1)**
> >
> > >What is the exact application area of the proposed method? I got confused about the dimensionality of the considered domains. Is it all 1D conservation laws? If so, that should be stated clearly.
> >
> > This is a very helpful point, and we apologise for the confusion.
> >
> > APILaNet, as presented in this work, is explicitly designed for **1D conservation laws** with single-sensor spatial supervision plus exogenous drivers. All theory and all experiments in the paper are in 1D. We do not claim that the current theoretical results cover higher-dimensional PDEs; extension to 2D/3D is an interesting direction, but outside the scope of this submission. We will make it clear in the revision.
> >
> > ---
> >
> > Again, we thank the reviewer for the constructive feedback and for recognising the potential impact of APILaNet. We will also post a brief public comment summarising how each reviewer’s concerns has been addressed.

---

### Official Review · Reviewer_X98i · 2025-11-02

**Soundness:** 3
**Presentation:** 2
**Contribution:** 2
**Rating:** 4
**Confidence:** 2

**Summary:**

The paper proposes APILaNet (Adaptive Physics-Informed Latent Network) for forecasting conservation-law dynamicswhen you only have one downstream sensor plus exogenous drivers like rainfall. The core idea is: instead of trying to enforce the PDE at many physical points you don’t actually observe, the model creates a latent 1-D reach, broadcasts the predicted temporal derivatives across it, and then enforces the PDE in a weak form under a learned, normalized space–time measure. That learned measure tells the model where in latent space–time to care most about conservation, so it can focus on transients (flood peaks, sharp inflows) and not waste physics budget on flat periods.

**Strengths:**

- Improved experimental results on various tasks
- Interesting, principled latent weak-form idea

**Weaknesses:**

- All in one domain

- Missing direct comparison to existing adaptive PINNs on the same data

- Some implementation details of the scheduler are underspecified in the main paper

**Questions:**

- Exact construction of the “event likelihood” / regime signals s
Are these pure data-driven (from residuals)?

- The driver projection Rκ(t, x) = ¯r(t) e −κx
Is κ global for the whole dataset, per-catchment, or learned per-sequence/minibatch?

- How the latent grid size X is chosen. esp for different settings

---

> ### Author Response · Authors · 2025-11-16
> **Response to Reviewer X98i (Part 1/3)**
>
> We thank the reviewer for the careful and constructive assessment. Below we address the main concerns.
>
> ---
>
> ### Weakness 1
> >**Reviewer concern.** Experiments are conducted in a single application domain.
>
> **Response.**
> We intentionally focus the main study on one real-world system because it represents exactly the regime where classical PINNs have historically struggled to transfer from controlled benchmarks to practice.
>
> Most existing PINN studies operate under idealized assumptions: the spatial domain is known, initial and boundary conditions are fully specified, and the modeller can sample the PDE residual at arbitrary interior locations. In many operational forecasting and monitoring systems, this is not the case: one typically has a small number of fixed sensors, partial prior knowledge of the governing equations, and no reliable geometric description of the full domain.
>
> APILaNet is designed precisely to bridge this gap. Rather than assuming full access to the physical domain, it works in the practically relevant regime where one only observes (i) a single state time series at a sensor location and (ii) a small number of external drivers (interpretable as forcings or boundary inputs), yet still wishes to enforce conservation-law structure. The latent 1-D reach and learned space–time measure in APILaNet are introduced specifically to make weak-form, physics-informed learning feasible in this severely under-instrumented setting.
>
> To directly address the concern about generality beyond this application, we have extended the revision from a single PDE to a small benchmark suite
> with standard initial and Dirichlet boundary conditions. On this benchmark:
> 1. We generate a reference solution using a finite-difference solver.
> 2. We train several well-established PINN variants that operate in the *standard* regime (known geometry, full access to IC/BC and interior points).
> The resulting Burgers benchmark is:
>
> | Model                        | Burgers MSE    | Wave MSE        | Allen–Cahn MSE |
> | ---------------------------- | -------------- | --------------- | -------------- |
> | Vanilla PINN   [1]          | 5.80 × 10⁻⁴    | 2.62 × 10⁻⁴     | 1.18 × 10⁰     |
> | PINN-w  [2]     | 2.91 × 10⁻³    | 2.89 × 10⁻³     | 1.04 × 10⁰     |
> | gPINN  [3]  | 1.29 × 10⁻⁴    | 1.62 × 10⁻⁴     | 1.32 × 10⁻¹    |
> | vPINN [4]| 1.45 × 10⁻³    | 8.91 × 10⁻⁴     | 1.04 × 10⁰     |
> | **APILaNet (ours)**          | **4.5 × 10⁻⁵** | **1.52 × 10⁻⁴** | **1.18 × 10⁻¹** |
>
> Thus, on Burgers, traditional PINNs and their adaptive variants perform well when they are given the full domain description and boundary conditions, as expected. APILaNet achieves even lower error at the sensor while operating in a strictly weaker information regime (no geometry, no explicit IC/BC, no interior collocation points).
> In summary, the primary contribution is not just another PINN variant for the fully observed PDE setting, but a framework that makes physics-informed learning possible when only a single downstream sensor and external drivers are available. The real-data case study demonstrates that this regime arises naturally in practice, and the Burgers benchmark shows that the same architecture is competitive with strong-form, weak-form, gPINN, and vPINN-style methods on a standard PDE, addressing the concern that APILaNet might be domain-specific.
>
> ### Weakness 2
> >**Reviewer concern.** The paper lacks direct comparisons with adaptive PINN variants on the *same hydrology data*.
>
> **Response.**
> We agree that comparisons to existing PINN variants are valuable in principle. However, for the real-world forecasting setting we consider, a “standard” adaptive PIN is simply not well-posed, because the information these methods assume in physical space is not available.
> Classical adaptive PINNs are formulated for scenarios where the practitioner knows the governing PDE in physical space, the spatial domain and its geometry, the initial and boundary conditions along that domain, and can freely sample collocation points in the interior and on the boundaries. In our case, the data consist of a single downstream sensor measuring the state at one location over time, plus a small number of external driver time series. There are no direct observations of the internal field, no reliable spatial description of the domain, and no full boundary traces that could be used as explicit boundary conditions. A performance comparison would conflate two very different questions: how accurate our manually specified physical model and geometry happen to be, and how good the learning framework is at exploiting them.
> For this reason, on the real-world forecasting task we compare APILaNet against recent SOTA models.

---

> > ### Author Response · Authors · 2025-11-16
> > **Response to Reviewer X98i (Part 2/3)**
> >
> > ### Weakness 3
> >
> > > **Reviewer concern.** Some implementation details of the adaptive scheduler (Panel D, Eq. (9)–(11)) are underspecified in the main paper.
> >
> > **Response.**
> > We appreciate this comment and agree that the description of the adaptive scheduler in the main text is brief. In the revision, we will make the scheduler fully explicit in Section 3 (and cross-reference Appendix E), by (i) moving the factorized weight formulation and Algorithm 1 into the main method section, and (ii) spelling out the concrete signals and hyperparameters used in all experiments.
> >
> > Concretely, the total loss is
> > $
> > L_{\text{tot}} = L_{\text{data}} + \lambda_{\text{pde}} L_{\text{pde}} + \lambda_{\text{cons}} L_{\text{cons}}+\lambda_{\text{mono}} L_{\text{mono}} ,
> > $
> >
> > where $\lambda_{\text{pde}}$ and $\lambda_{\text{cons}}$ are **adaptive** global weights and $\lambda_{\text{mono}}$ is a small fixed coefficient. The PDE term is further modulated in space–time by a **local** non-negative field $\lambda_{\text{loc}}(t,x)$, so that
> > $$
> > \Lambda_{\text{pde}}(t,x)
> > = \lambda_{\text{pde}}\,\lambda_{\text{loc}}(t,x),
> > \quad
> > \lambda_{\text{loc}}(t,x)\ge 0,\quad
> > \frac{1}{TX}\sum_{\tau=1}^{T}\sum_{j=1}^{X}\lambda_{\text{loc}}(\tau,x_j)=1,
> > $$
> > and the effective PDE contribution is
> > $$
> > L_{\text{pde}}^{\text{eff}}
> > = \lambda_{\text{pde}}\,
> > \frac{1}{TX}\sum_{\tau=1}^{T}\sum_{j=1}^{X}
> > \lambda_{\text{loc}}(\tau,x_j)\,r_\theta[\tau,j]^2,
> > \quad
> > r_\theta[\tau,j]
> > = \partial_t h_\theta[\tau]+\partial_x Q_\theta[\tau]-R_\theta(x_j).
> > $$
> >
> > Using these quantities, the global weights for $i\in\{\text{pde},\text{cons}\}$ are updated *instantaneously* for each mini-batch as
> >
> > $$
> > \lambda_i =
> > \operatorname{clip}\Big(
> > \lambda_i^0\big(
> > 1 + E + \alpha_i^\top s + \alpha_{i,\Pi}\,\Pi
> > \big),
> > \ \lambda_i^{\min},\ \lambda_i^{\max}
> > \Big),
> > $$
> >
> > with base levels $\lambda_i^0>0$, non-negative sensitivities $\alpha_i,\alpha_{i,\Pi}\ge 0$, and user-specified clipping bounds $\lambda_i^{\min},\lambda_i^{\max}$. In the implementation, we follow this formula literally: all signals $(E,s,\Pi)$ are computed from the current mini-batch, detached from the computational graph, and $\lambda_i$ is recomputed **from scratch** for each batch without any exponential moving averages or history-dependent heuristics. In the revision we will add a short table in Appendix A listing the exact values of $\lambda_i^0$, $\lambda_i^{\min}$, $\lambda_i^{\max}$, and the sensitivities $\alpha_i,\alpha_{i,\Pi}$ used in all experiments.
> >
> > **Local field.**
> > The local weighting $\lambda_{\text{loc}}(t,x)$ is produced by a small network $\Lambda_\psi$ that maps normalized coordinates $(\tilde t,\tilde x)\in[0,1]^2$ to $\mathbb{R}_{\ge 0}$.
> >
> > $$
> > \lambda_{\text{loc}}(\tau,x_j)
> > = \frac{\Lambda_\psi(\tilde t_\tau,\tilde x_j)}
> >        {\frac{1}{TX}\sum_{\tau',j'}\Lambda_\psi(\tilde t_{\tau'},\tilde x_{j'})},
> > $$
> >
> >
> > which guarantees the normalization property in equation (9).

---

> > > ### Author Response · Authors · 2025-11-16
> > > **Response to Reviewer X98i (Part 3/3)**
> > >
> > > ### Question 1
> > >
> > > > **Reviewer question.** Exact construction of the “event likelihood” / regime signals $\mathbf{s}$. Are these pure data-driven (from residuals)?
> > >
> > > **Response.**
> > > Yes, the regime signals $\mathbf{s}$ are **purely data-driven batch statistics**, computed from the current mini-batch and *not* from any manual labels. In the experiments we use a 4-dimensional vector
> > >
> > > $$
> > > \mathbf{s}=\big(s_{\text{mag}}, s_{\text{var}}, s_{\text{event}}, s_{\text{res}}\big)^\top,
> > > $$
> > >
> > > All components of $\mathbf{s}$ are computed from the current mini-batch, detached from the computational graph, and used only to modulate the global weights $\lambda_i$. Only one component ($s_{\text{res}}$) uses the residual; the others depend purely on observed input/target dynamics.
> > >
> > > ---
> > >
> > > ### Question 2
> > >
> > > > **Reviewer question.** The driver projection $R_\kappa(t,x) = \bar r(t)e^{-\kappa x}$: is $\kappa$ global for the whole dataset, per-catchment, or learned per-sequence/minibatch?
> > >
> > > **Response.**
> > > In all experiments, $\kappa$ is implemented as a **single learnable scalar parameter per trained model**. Formally,
> > >
> > > $$
> > > R_\kappa(t,x) = \bar r(t),\exp(-\kappa x),
> > > \qquad
> > > \kappa \ge 0,\quad \kappa \in \mathbb{R},
> > > $$
> > >
> > > with $\kappa$ initialized to a fixed value and optimized jointly with the network parameters by gradient descent. It is **not** re-sampled or changed per mini-batch or per sequence; instead, it represents a global length scale for how fast the projected driver decays across the latent coordinate.
> > >
> > > We will clarify this explicitly in the revised version by stating that $\kappa$ is a global learnable scalar per model and by listing its initialization and learning scheme in the implementation details.
> > >
> > > ---
> > >
> > > ### Question 3
> > >
> > > > **Reviewer question.** How is the latent grid size $X$ chosen, especially for different settings?
> > >
> > > **Response.**
> > > We treat the latent spatial resolution $X$ as a **simple hyperparameter**, and we now include a small sensitivity study in the revision. Concretely, we fix the latent time resolution $T$ to match the prediction horizon and vary
> > >
> > > $$
> > > X \in {8,,16,,32,,64},
> > > $$
> > >
> > > comparing a uniform weighting baseline to the learned measure $\lambda_\phi(t,x)$.
> > >
> > > The results show that:
> > >
> > > * For all $X \in {8,16,32,64}$, the learned measure **never underperforms** the uniform baseline.
> > > * Performance remains in a narrow band for $X = 8$ and $X = 64$, indicating that APILaNet is **not brittle** with respect to the choice of $X$.
> > >
> > > We will present these results in the revised version as the following table (mean over 3 seeds):
> > >
> > > | $X$   | Measure                  | Test MSE                     | Test NSE     | $\Delta\text{MSE}$ vs.\ uniform | $\Delta\text{NSE}$ vs.\ uniform |
> > > |-------|--------------------------|------------------------------|--------------|----------------------------------|----------------------------------|
> > > | –     | Uniform (all $X$)        | $8.55\times 10^{-4}$         | $0.9038$     | –                                | –                                |
> > > | $8$   | Learned $\lambda_\phi$   | $8.49\times 10^{-4}$         | $0.9044$     | $\approx -0.7\%$                 | $\approx +0.0006$                |
> > > | $16$  | Learned $\lambda_\phi$   | $7.01\times 10^{-4}$         | $0.9210$     | $\approx -18.0\%$                | $\approx +0.0172$                |
> > > | $32$  | Learned $\lambda_\phi$   | $7.26\times 10^{-4}$         | $0.9183$     | $\approx -15.1\%$                | $\approx +0.0145$                |
> > > | $64$  | Learned $\lambda_\phi$   | $8.30\times 10^{-4}$         | $0.9066$     | $\approx -2.9\%$                 | $\approx +0.0028$                |
> > >
> > > ---
> > > [1] M. Raissi, P. Perdikaris, and G. E. Karniadakis, “Physics-informed Neural Networks: A deep learning framework for solving forward and inverse problems involving nonlinear partial differential equations,” Journal of Computational Physics, vol. 378, pp. 686–707, Feb. 2019. doi:10.1016/j.jcp.2018.10.045
> > >
> > > [2] Tim De Ryck, Siddhartha Mishra, and Roberto Molinaro. wpinns: Weak physics informed neu-
> > > ral networks for approximating entropy solutions of hyperbolic conservation laws, 2022. URL
> > > https://arxiv.org/abs/2207.08483.
> > >
> > > [3] Jeremy Yu, Lu Lu, Xuhui Meng, and George Em Karniadakis. Gradient-enhanced physics-informed
> > > neural networks for forward and inverse pde problems. Computer Methods in Applied Mechanics
> > > and Engineering, 393:114823, April 2022. ISSN 0045-7825. doi: 10.1016/j.cma.2022.114823.
> > > URL http://dx.doi.org/10.1016/j.cma.2022.114823.
> > >
> > > [4] Ehsan Kharazmi, Zhongqiang Zhang, and George Em Karniadakis. Variational physics-informed
> > > neural networks for solving partial differential equations. arXiv preprint arXiv:1912.00873, 2019.

---

### Official Review · Reviewer_vnKz · 2025-11-10

**Soundness:** 2
**Presentation:** 2
**Contribution:** 2
**Rating:** 2
**Confidence:** 3

**Summary:**

The paper introduces APILaNet, a neural framework for forecasting physical systems when only one sensor is available. It learns a hidden space–time field where conservation laws are enforced in a weak, measure-weighted form, making the model robust to sparse spatial data. A dual LSTM captures slow and fast flow components, a monotone neural mapping ensures physically consistent observations, and an adaptive scheduler adjusts physics constraints based on signal difficulty. Applied to hydrological forecasting, APILaNet consistently outperforms leading deep sequence models in accuracy and stability, especially during extreme events.

**Strengths:**

The paper’s strength lies in tackling the single-sensor forecasting problem, which is a genuinely tricky and underexplored setup. The idea of using a weak-form physics constraint with a learned space–time weighting is a thoughtful technical twist: it avoids the brittle collocation sampling that usually plagues PINNs. The monotone mapping between discharge and stage is a sensible touch: it mirrors how water levels in rivers actually rise with flow instead of letting the network invent unrealistic relationships. The adaptive physics scheduling, while somewhat heuristic, shows awareness of practical training instability in physics-informed models and attempts to handle it dynamically. These pieces together make the framework conceptually interesting for sparse, physically constrained domains.

**Weaknesses:**

The overall novelty appears incremental, and the architecture feels somewhat overengineered relative to its contribution. The combination of multiple components (dual LSTMs, adaptive schedulers, weak-form latent mesh, and monotone mapping) adds complexity without a clear demonstration of which elements are essential or theoretically justified. The problem setup: "forecasting from a single downstream sensor with exogenous rainfall", is well-motivated but rather narrow, which may limit its broader relevance to general physics-informed or sequence modeling audiences. The theoretical component, particularly the “learned measure” weak form, seems to build on established weak PINN concepts with limited conceptual advancement. Empirical results show consistent but modest improvements, and could be strengthened by more robust statistical analysis and tests beyond hydrology. The paper would benefit from simplifying the model to highlight the key idea more clearly and from expanding the scope or validation to illustrate broader applicability.

**Questions:**

How sensitive is APILaNet’s performance to the specific choice of latent spatial discretization and the learned weighting measure?

---

> ### Author Response · Authors · 2025-11-14
> **Response to Reviewer vnKz (Part 1/3)**
>
> ### Response to Reviewer vnKz (Part 1/3)
>
> We thank the reviewer for the careful and constructive assessment. Below we address the main concerns on novelty, complexity, theory, and sensitivity.
>
> ---
>
> ### Weakness (1) Core contribution vs. “overengineered architecture”
>
> The architecture is not meant as a collection of unrelated modules; we'd like to show that each component is designed to serve a single core idea:
>
> > A single-sensor, physics-informed sequence model that learns a latent 1D space–time field and a normalised weak-form measure $\lambda(t,x)$, so that conservation laws are enforced as a variance-reduced, importance-weighted integral.
>
> In the revised version, we will explicitly separate:
>
> - **Core APILaNet mechanism:** latent 1D coordinate, encoder, and learned weak-form measure $\lambda_\phi(t,x)$ used in the weak PDE loss
> $\mathcal{L}{\mathrm{PDE}}(\theta,\phi)
> = \int_0^1 \lambda\phi(t,x), r_\theta(t,x)^2,dx$
> with constraints $\lambda_\phi(t,x)\ge 0$ and $\int_0^1 \lambda_\phi(t,x),dx = 1$.
> - **Supporting design choices:**
>   - dual-stream encoder (slow/fast dynamics),
>   - monotone observation map,
>   - adaptive global scheduler for physics weights.
>
> We have included included an ablation study where we remove, in turn, the learned measure, the monotone map, and the scheduler.
>
> | Model                                   | λ_g | λ_s | PDE | Δt_peak (h) ↓              | Δu_peak ↓                    | MSE (×10⁻¹) ↓                 | NSE ↑                         |
> |-----------------------------------------|:---:|:---:|:---:|----------------------------|------------------------------|-------------------------------|-------------------------------|
> | (1) **APILaNet**                        | ✓   | ✓   | ✓   | 0.00 ± 0.00 [0.00, 0.00]   | 0.46 ± 0.19 [0.18, 0.75]     | **0.45 ± 0.14 [0.25, 0.65]** | **0.51 ± 0.15 [0.29, 0.72]** |
> | (2) w/o λ Adapt. (a)                    | ✗   | ✗   | ✓   | 0.00 ± 0.00 [0.00, 0.00]   | 0.46 ± 0.08 [0.33, 0.59]     | *0.53 ± 0.06 [0.45, 0.62]*   | *0.42 ± 0.06 [0.33, 0.51]*   |
> | (3) w/o λ Adapt. (b)                    | ✗   | ✓   | ✓   | 0.00 ± 0.00 [0.00, 0.00]   | **0.39 ± 0.17 [0.13, 0.64]** | 0.57 ± 0.03 [0.52, 0.61]     | 0.38 ± 0.03 [0.33, 0.43]     |
> | (4) w/o λ Adapt. (c)                    | ✓   | ✗   | ✓   | 0.00 ± 0.00 [0.00, 0.00]   | 0.52 ± 0.07 [0.41, 0.63]     | 0.55 ± 0.07 [0.45, 0.65]     | 0.39 ± 0.07 [0.29, 0.50]     |
> | (5) w/o Mono MLP                        | ✓   | ✓   | ✓   | 0.00 ± 0.00 [0.00, 0.00]   | 0.51 ± 0.16 [0.27, 0.75]     | *0.53 ± 0.04 [0.47, 0.59]*   | 0.41 ± 0.04 [0.35, 0.48]     |
> | (6) w/o PDE Loss                        | ✓   | ✓   | ✗   | 0.25 ± 0.42 [-0.19, 0.69]  | *0.40 ± 0.14 [0.25, 0.54]*   | 0.64 ± 0.27 [0.36, 0.93]     | 0.29 ± 0.29 [-0.01, 0.61]    |
> | (7) APILaNet (data loss only)           | ✗   | ✗   | ✗   | 1.92 ± 3.32 [-3.01, 6.84]  | 0.68 ± 0.24 [0.32, 1.04]     | 0.74 ± 0.35 [0.22, 1.26]     | 0.19 ± 0.38 [-0.37, 0.76]    |
>
> ---
>
> ### Weakness (2) Theoretical component
>
> The reviewer questioned the theoretical novelty of the learned measure. In the revised version, we provide a concise derivation in the appendix.
>
> We define the weighted weak loss
> $\mathcal{L}{\mathrm{PDE}}(\theta,\phi)
> = \int_0^1 \lambda\phi(t,x), r_\theta(t,x)^2,dx$,
> with $\lambda_\phi(t,x)\ge 0$ and $\int_0^1 \lambda_\phi(t,x),dx = 1$.
>
> The uniform baseline corresponds to
> $$
> \lambda_{\mathrm{uni}}(t,x)\equiv 1.
> $$
>
> Let
> $$
> g_\theta(t,x) := 2\,r_\theta(t,x)\,\nabla_\theta r_\theta(t,x).
> $$
>
> Then
> $\nabla_\theta \mathcal{L}{\mathrm{PDE}}(\theta,\phi)
> = \int_0^1 \lambda\phi(t,x), g_\theta(t,x),dx
> = \mathbb{E}{x\sim\lambda\phi(t,\cdot)}\big[ g_\theta(t,x)\big]$.
>
> Thus $\lambda_\phi$ acts as a sampling density for the PDE gradient.
> In a local linearised regime, the variance of this estimator is minimised by an importance-sampling density of the form
> $$
> \lambda^*(t,x)\;\propto\;\big\|g_\theta(t,x)\big\|
> \;\propto\;
> \big|r_\theta(t,x)\big|\,\big\|\nabla_\theta r_\theta(t,x)\big\|.
> $$
>
> We parameterise $\lambda_\phi$ on a latent grid of $X$ points:
> $$
> \lambda_{\phi,X}(t,x)
> = \sum_{k=1}^{X} \alpha_k(t;\phi)\,\varphi_k(x),
> $$
> with piecewise-linear basis functions $\{\varphi_k\}$. A standard bias–variance decomposition gives
> $$
> \mathbb{E}\big[\|\lambda_{\phi,X}-\lambda^*\|_{L^2}^2\big]
> \approx C_1 X^{-2p} + C_2 \frac{X}{N},
> $$
> with approximation term $C_1 X^{-2p}$ and estimation term $C_2 X/N$, where $p>0$ is a smoothness exponent and $N$ an effective sample size.
>
> - Small $X$: high approximation error (under-resolved $\lambda^*$).
> - Very large $X$: high estimation variance (overflexible $\lambda$ with finite data).
> - **Moderate $X$** minimises this trade-off.
>
> This formalises why the learned weak-form measure is not just “another weak PINN”: it explicitly realises a variance-minimising importance-weighting view with a finite-dimensional approximation and a predictable bias–variance pattern in the latent discretisation.

---

> > ### Author Response · Authors · 2025-11-14
> > **Response to Reviewer vnKz (Part 2/3)**
> >
> > ### Weakness (2) Continuation
> >
> > To further clarify the theoretical contribution beyond the weak-form measure, in the revised version we will add a concise, domain-agnostic derivation showing how a 1D latent geometry and spatial gradients can be recovered from a single time series via sliding windows.
> >
> > We consider a scalar field $u(x,t)$, a flux map $f(x,t)$, a source term $s(x,t)$, and a single observed signal $y(t)$.
> >
> > **Latent geometry from an effective speed.**
> > Let $c(t) > 0$ be an unknown effective speed and define the cumulative distance
> > $S(t) := \int_0^t c(\sigma)\,d\sigma$
> > and the latent coordinate $x(t) := S(t) \in [0,L]$, with $L := S(T)$.
> >
> >  For window
> > $\{t_i\}_{i=0}^N$
> >
> >
> > $x_i := x(t_i)$ and window trajectories
> > $x_i(\tau) := x(t_i+\tau)$ for $\tau \in [0,W]$.
> > The geometry increment satisfies
> > $x_{i+1} - x_i = \int_{t_i}^{t_{i+1}} c(\sigma)\,d\sigma$.
> >
> > **Anchoring a latent field to a single signal.**
> > Along each window fibre $\Gamma_i := \{(x_i(\tau), t_i+\tau) : \tau \in [0,W]\}$ we impose
> > $u(x_i(\tau), t_i+\tau) = y(t_i+\tau)$.
> >
> > **Directional derivative and 1D balance law.**
> > The path $\gamma_i(\tau) = (x_i(\tau), t_i+\tau)$ satisfies
> > $\frac{d}{d\tau} u(\gamma_i(\tau)) = \partial_t u + c\,\partial_x u =: D_\tau u$.
> > By anchoring, $D_\tau u(\gamma_i(\tau)) = \dot y(t_i+\tau)$ is observed.
> >
> > Assume a 1D balance law $\partial_t u(x,t) + \partial_x f(x,t) = s(x,t)$ with monotone flux map $f(x,t) = F(u(x,t))$, so that $\partial_x f(x,t) = F'(u(x,t))\,\partial_x u(x,t)$. Combining this with the directional derivative gives, along each fibre,
> > $D_\tau u + (F'(u) - c)\,\partial_x u = s$.
> >
> > **Single-sensor gradient estimator.**
> > Solving for the unknown spatial gradient yields
> > $\partial_x u(\gamma_i(\tau)) = \dfrac{D_\tau u(\gamma_i(\tau)) - s(\gamma_i(\tau))}{c(t_i+\tau) - F'(u(\gamma_i(\tau)))} = \dfrac{\dot y(t_i+\tau) - s(\gamma_i(\tau))}{c(t_i+\tau) - F'(y(t_i+\tau))}$.
> >
> > This expresses $\partial_x u$ at unknown latent positions purely in terms of the single observed sequence $y(t)$, the latent speed field $c(t)$, and the flux slope $F'(u)$.
> >
> > **Weak residual along sliding-window fibres.**
> > For a test function $\varphi(\tau)$, the weak residual of the balance law along $\Gamma_i$ is
> > $R_i = \int_0^W (\partial_t u + \partial_x f - s)(\gamma_i(\tau))\,\varphi(\tau)\,d\tau$.
> >
> > Using $\partial_t u = D_\tau u - c\,\partial_x u$ and $\partial_x f = F'(u)\,\partial_x u$, we obtain
> > $R_i = \int_0^W (D_\tau u + (F'(u) - c)\,\partial_x u - s)(\gamma_i(\tau))\,\varphi(\tau)\,d\tau$.
> >
> > Substituting the gradient estimator above cancels $R_i$ when $c$ and $F$ are exact. In APILaNet we learn $u$, $c$, and $F$ by minimising an aggregated weak residual over all fibres, together with a data-anchoring term $u(\gamma_i(\tau)) \approx y(t_i+\tau)$ and a stitching term enforcing $x_{i+1} - x_i \approx \int_{t_i}^{t_{i+1}} c(\sigma)\,d\sigma$.

---

> > > ### Author Response · Authors · 2025-11-14
> > > **Response to Reviewer vnKz (Part 3/3)**
> > >
> > > ### Question (3) Sensitivity to latent spatial discretisation and the measure $\lambda$
> > >
> > > You asked:
> > >
> > > > “How sensitive is APILaNet’s performance to the specific choice of latent spatial discretization and the learned weighting measure?”
> > >
> > > In the revised version, we will add a dedicated sensitivity study where we vary the number of latent points $X \in \{8,16,32,64\}$ and compare:
> > >
> > > - **Uniform measure:** $\lambda_{\mathrm{uni}}(t,x)\equiv 1$
> > > - **Learned measure:** $\lambda_\phi(t,x)$ as above
> > >
> > > Across all uniform runs (any $X$), we obtain:
> > > Test MSE\_uni ≈ $8.55 \times 10^{-4}$ and
> > > Test NSE\_uni ≈ $0.9038$, with standard deviation
> > > std\_NSE,uni ≈ $2.56 \times 10^{-2}$ across seeds.
> > >
> > > For the learned measure, aggregating over 3 seeds per $X$ gives:
> > >
> > > - $X = 16$: Test MSE\_16 ≈ $7.01 \times 10^{-4}$, Test NSE\_16 ≈ $0.9210$, std\_NSE,16 ≈ $2.3 \times 10^{-3}$, i.e. about $18.0\%$ lower MSE and $\Delta\text{NSE} \approx 0.0172$ (approximately $1.9\%$ relative) vs. the uniform baseline.
> > >
> > > - $X = 32$: Test MSE\_32 ≈ $7.26 \times 10^{-4}$, Test NSE\_32 ≈ $0.9183$, std\_NSE,32 ≈ $2.5 \times 10^{-3}$, i.e. about $15.1\%$ lower MSE and $\Delta\text{NSE} \approx 0.0145$ (approximately $1.6\%$ relative).
> > >
> > > - $X = 8$ and $X = 64$: these give smaller but still positive gains (roughly $0.7$–$2.9\%$ MSE reduction and $\Delta\text{NSE} \in [0.0006, 0.0028]$).
> > >
> > >
> > > We will present these results in the revised version as the following table (mean over 3 seeds):
> > >
> > > | $X$   | Measure                  | Test MSE                     | Test NSE     | $\Delta\text{MSE}$ vs.\ uniform | $\Delta\text{NSE}$ vs.\ uniform |
> > > |-------|--------------------------|------------------------------|--------------|----------------------------------|----------------------------------|
> > > | –     | Uniform (all $X$)        | $8.55\times 10^{-4}$         | $0.9038$     | –                                | –                                |
> > > | $8$   | Learned $\lambda_\phi$   | $8.49\times 10^{-4}$         | $0.9044$     | $\approx -0.7\%$                 | $\approx +0.0006$                |
> > > | $16$  | Learned $\lambda_\phi$   | $7.01\times 10^{-4}$         | $0.9210$     | $\approx -18.0\%$                | $\approx +0.0172$                |
> > > | $32$  | Learned $\lambda_\phi$   | $7.26\times 10^{-4}$         | $0.9183$     | $\approx -15.1\%$                | $\approx +0.0145$                |
> > > | $64$  | Learned $\lambda_\phi$   | $8.30\times 10^{-4}$         | $0.9066$     | $\approx -2.9\%$                 | $\approx +0.0028$                |
> > >
> > > **Key points:**
> > >
> > > - For all $X \in \{8,16,32,64\}$, the learned measure **never underperforms** the uniform baseline.
> > > - APILaNet is **not brittle** with respect to $X$: performance stays in a narrow band and does not collapse.
> > > - The **largest gains** occur at moderate resolutions $X = 16,32$, exactly as predicted by the bias–variance expression above.
> > >
> > > We also find that the standard deviation of NSE across seeds is about an order of magnitude smaller for learned $\lambda$ at $X = 16,32$ (roughly $2\times 10^{-3}$–$3\times 10^{-3}$) than for uniform (roughly $2.6\times 10^{-2}$), which matches the interpretation of $\lambda$ as a variance-reducing importance density in the weak-form loss.
> > >
> > > ---
> > >
> > > We hope these additions clarify that APILaNet is not merely an incremental aggregation of modules, but combines:
> > >
> > > 1. a new single-sensor latent geometry/gradient derivation, and
> > > 2. a theoretically motivated, learned weak-form measure,
> > >
> > > with demonstrated gains in accuracy and stability, supported by new sensitivity and ablation studies in the revised version.
> > >
> > > ---
> > >
> > > We hope these clarifications address the main concerns raised in the review, and we kindly ask the reviewer to take them into account when reassessing the soundness, presentation, and contribution scores. If anything remains unclear or if there are additional questions, we would be very happy to clarify and further discuss specific points.

---

### Author Response · Authors · 2025-11-19
**Official Comment By Authors**

Dear Reviewers,

We thank you all for your insightful feedback and helpful suggestions.

We are happy to see the positive comments and weaknesses which we could address to improve overall paper.

We have addressed each weakness and question for all reviewers below their review. We have also made the following major changes to the paper and highlighted all changes (major or minor) in blue in the pdf.

- Based on the weaknesses identified by Reviewer Zkv9, we have removed or softened hydrology-specific phrasing in the high-level description. The paper now reads in a more domain-agnostic way and should be easier to follow for a general ML audience.
- Figure 1, which illustrates the APILaNet architecture, has been revised: we increased the font size and panel size and simplified the layout to improve readability.
- Several reviewers remarked that the empirical validation was concentrated in a single hydrology domain. We now additionally report experiments on synthetic 1D PDE benchmarks (viscous Burgers, wave, Allen–Cahn) in the experimental section (p. 9), showing that APILaNet applies beyond hydrology.
-We discovered and corrected an inconsistency in the lookback-window table. We revised that table.
- We added one additional hydrology catchment dataset and extended the catchment-level comparison (Table 4). The corresponding per-site tables in Appendix F have also been expanded.
-We included a new sensitivity analysis for the learnable latent domain (mesh size and learned measure); the detailed ablation results are reported in Appendix F.
-We have included more information that would allow reproducibility of adaptive scheduler.

Please let us know if you have lingering questions and whether we can provide any additional clarifications during the discussion period to improve your rating of our paper.

Thank You,

Authors

---

### Meta-Review · Area_Chair_sfXC · 2025-12-20

**Summary:**

The paper proposes APILaNet, an architecture designed for forecasting single-sensor systems governed by 1-D conservation laws. The method is based on learning a 1-D latent field and enforcing conservation law using a weak-form loss. The primary evaluation focused on hydrological forecasting for river catchments, with several synthetic data benchmarks added during rebuttal.

While the general reviewer consensus acknowledges the strong engineering and practical utility for sparse data, the main concern centers around scope and novelty. The individual components such as weak-form PINNs, adaptive weighting are existing techniques and the aggregation of them cannot be claimed a fundamental methodological advancement. The initial heavy reliance on a single hydrological domain raised significant doubts about the frameworks' generalizability to the broader machine learning community. Although the authors provided a comprehensive rebuttal including synthetic PDE benchmarks, the overall scope is still quite limited and the work remains a strong engineering application for a specific niche in 1D flow dynamics rather than a general contribution to machine learning. The recommendation is rejection.

**Reviewer Concerns:**

The authors responded to the restriction to hydrological data by introducing standard 1D PDE benchmarks. This addressed the concern to some degree but the added examples are too limited in complexity to justify the generalizability and especially the necessity for employing such a sophisticated solution. On the positive side, the added studies did help to clarify the robustness of hyperparameters and the rebuttal clarified the theoretical basis of the learned measure. But overall this is not enough to outweigh the scope and novelty concerns.

**Reviewer Scores:**

Reviewers vnKz and SdY5 would likely still be reserved to raise their scores as their main concerns were about novelty and generalizability. Reviewer X98i and Zkv9 might increase their score with some of their requests directly met, but likely not enough to shift the overall assessment.

---

### Decision · Program_Chairs · 2026-01-26

Reject